# Assessment of Salvage Surgery in Persistent Cervical Cancer after Definitive Radiochemotherapy: A Systematic Review

**DOI:** 10.3390/medicina59020192

**Published:** 2023-01-18

**Authors:** Carmine Conte, Luigi Della Corte, Silvia Pelligra, Giuseppe Bifulco, Biagio Abate, Gaetano Riemma, Marco Palumbo, Stefano Cianci, Alfredo Ercoli

**Affiliations:** 1Department of General Surgery and Medical-Surgical Specialties, Institute of Obstetrics and Ginecology, A.O.U. Policlinico Rodolico—San Marco, University of Catania, 95125 Catania, Italy; 2Department of Neuroscience, Reproductive Sciences and Dentistry, School of Medicine, University of Naples Federico II, 80131 Naples, Italy; 3Department of Woman and Child Health and Public Health, Catholic University of the Sacred Heart, 00168 Rome, Italy; 4Department of Public Health, University of Naples Federico II, 80131 Naples, Italy; 5Department of Woman, Child and General and Specialized Surgery, Luigi Vanvitelli University of Campania, 81100 Naples, Italy; 6Unit of Gynecology and Obstetrics, Department of Human Pathology of Adult and Childhood “G. Barresi”, University of Messina, 98121 Messina, Italy

**Keywords:** locally advanced cervical cancer, concurrent chemoradiotherapy, salvage surgery, salvage hysterectomy, pelvic exenteration

## Abstract

*Background and Objectives:* The standard treatment approach in locally advanced cervical cancer (LACC) is exclusive concurrent chemoradiation therapy (RTCT). The risk of local residual disease after six months from RTCT is about 20–30%. It is directly related to relapse risk and poor survival, such as in patients with recurrent cervical cancer. This systematic review aims to describe studies investigating salvage surgery’s role in persistent/recurrent disease in LACC patients who underwent definitive RTCT. *Materials and Methods:* Studies were eligible for inclusion when patients had LACC with radiologically suspected or histologically confirmed residual disease after definitive RTCT, diagnosed with post-treatment radiological workup or biopsy. Information on complications after salvage surgery and survival outcomes had to be reported. The methodological quality of the articles was independently assessed by two researchers with the Newcastle–Ottawa scale. Following the recommendations in the Preferred Reporting Items for Systematic Reviews and Meta-Analyses (PRISMA) statement, we systematically searched the PubMed, Scopus, Cochrane, Medline, and Medscape databases in May 2022. We applied no language or geographical restrictions but considered only English studies. We included studies containing data about postoperative complications and survival outcomes. *Results:* Eleven studies fulfilled the inclusion criteria and all were retrospective observational studies. A total of 601 patients were analyzed concerning the salvage surgery in LACC patients for persistent/recurrent disease after RTCT treatment. Overall, 369 (61.4%) and 232 (38.6%) patients underwent a salvage hysterectomy (extrafascial or radical) and pelvic exenteration (anterior, posterior, or total), respectively. Four hundred and thirty-nine (73%) patients had histologically confirmed the residual disease in the salvage surgical specimen, and 109 patients had positive margins (overall range 0–43% of the patients). The risk of severe (grade ≥ 3) postoperative complications after salvage surgery is 29.8% (range 5–57.5%). After a median follow-up of 38 months, the overall RR was about 32% with an overall death rate of 40% after hysterectomy or pelvic exenteration with or without lymphadenectomy. *Conclusions:* There is heterogeneity between the studies both in their design and results, therefore the effect of salvage surgery on survival and recurrence cannot be adequately estimated. Future homogeneous studies with an appropriately selected population are needed to analyze the safety and efficacy of salvage hysterectomy or pelvic exenteration in patients with residual tumors after definitive RTCT.

## 1. Introduction

Cervical cancer represents one of the most common malignancies worldwide and the second most common cause of death in females. In Europe, the annual incidence is around 60,000, with a mortality rate of around 30,000 [1].

Prognosis is favorable in early-stage disease (FIGO stage IA, B1-2, IIA1) with a 5-year survival rate of around 80–90%; conversely, patients bearing locally advanced (FIGO stage IB3-IVA) or metastatic disease (FIGO stage IVB) experienced a 5-year survival rate of 70% and 15–20%, respectively [2].

The standard treatment approach in locally advanced cervical cancer (LACC) (stage IB3–IVA) is concurrent chemoradiation therapy (RTCT), which generally consists of cisplatin-based chemotherapy and external-beam radiotherapy (EBRT), followed by intracavitary brachytherapy (ICBT) based on results from randomized clinical trials demonstrating the superiority of exclusive RTCT over radiotherapy only (73% versus 52%, respectively) [3,4,5].

According to international guidelines, the treatment algorithm should aim to avoid the combination of RTCT and radical hysterectomy and pelvic exenteration (PE) in LACC patients due to the significant increase in morbidity and no evident impact on survival [6,7].

Even if definitive RTCT has significantly improved tumor control [2], local failure is still seen in the range of 10–40% according to FIGO stage (IB3-IVA), and a small group of patients can be unresponsive to definitive RTCT and brachytherapy [8,9].

The residual disease after 6 months from RTCT is considered microscopic (persistent tumor foci ≤3 mm maximum dimension) (pmicroR), or macroscopic (persistent tumor foci >3 mm maximum dimension) (pmacroR), according to pathology [10].

The recurrent disease is defined as the clinical or pathological recurrence after a complete response at least six months later the RTCT [7].

In a randomized trial, Morice et al. suggested that salvage hysterectomy had no therapeutic impact in patients with clinical and radiological complete response after RTCT, but this conclusion is limited by the lack of power of the trial [11].

The risk of local residual disease after 6 months from RTCT is about 20–30%, and it is directly related to relapse risk and poor survival, such as in patients with recurrent cervical cancer [9].

In this patient population with persistent disease, pelvic local control remains the major issue, but there are no strong recommendations or evidence for optimal treatment.

A possible option is salvage surgery (SS). Simple or radical hysterectomy could be considered in carefully recurrent or persistent selected patients with small (<2 cm) lesions [7]. Instead, for cervical cancers where disease is confined to the central pelvis without distant metastasis, PE may be a potentially curative surgical option [7].

Moreover, when surgery is not feasible, another treatment approach is adjuvant chemotherapy, but its role is still unclear.

This systematic review aims to describe studies that investigate the role of salvage surgery for persistent/recurrent disease in LACC patients who underwent definitive RTCT.

Therefore, we analyzed the recurrence rate (RR), disease-free survival (DFS), overall survival (OS), and the perioperative outcomes in both salvage hysterectomy and PE.

## 2. Material and Methods

The methods for this study were specified a priori based on the recommendations in the Preferred Reporting Items for Systematic Reviews and Meta-Analyses (PRISMA) statement [12]. We obtained registration to the PROSPERO site for a systematic review.

### 2.1. Search Method

We systematically searched for studies about the addiction to hysterectomy (H) or pelvic exenteration (PE) in LACC patients who underwent definitive RTCT with persistent disease in the PubMed, Cochrane, Medline, and Medscape databases up to May 2022. We included the studies from the earliest publication reported in the scientific literature with no restriction on the country. We considered only English entirely published studies.

### 2.2. Study Selection

Study selection was made independently by CC and LDC. In case of discrepancy, a third person (SC) was questioned about its inclusion or exclusion. Inclusion criteria were: (1) studies that included patients with advanced FIGO stage cervical carcinoma (at least IB2); (2) studies that reported at least one outcome of interest (RR, OS, DFS); (3) peer-reviewed articles published initially. We excluded non-original studies, preclinical trials, animal trials, abstract-only publications, and articles in a language other than English. The studies selected and all reasons for exclusion are mentioned in the Preferred Reporting Items for Systematic Reviews and Meta-Analyses (PRISMA) flowchart (Figure 1). All included studies were assessed regarding potential conflicts of interest.

### 2.3. Statistical Analysis

The risk rate (RR) and 95% confidence intervals (CI) were used for dichotomous variables.

RR, DFS, and OS were used as clinical outcomes. In each study, the recurrence rate was defined as the percentage of relapse after the primary treatment (salvage surgery). Disease-free survival was defined as the time elapsed between surgery and recurrence or the date of the last follow-up. Overall survival was defined as the time elapsed between surgery and death or the date of the last follow-up.

### 2.4. Quality Assessment

The quality of the included studies was assessed using the Newcastle–Ottawa scale (NOS) [13]. This assessment scale uses three broad factors (selection, comparability, and exposure), with the scores ranging from 0 (lowest quality) to 8 (best quality). Two authors (CC, LDC) independently rated the study’s quality. Any disagreement was subsequently resolved by discussion or consultation with SC. The NOS scale is reported in Table 1.

## 3. Results

### 3.1. Studies’ Characteristics

Figure 1 illustrates the selection of studies for inclusion in the systematic review. From the bibliographic search, a total of 55 articles were retrieved. Fifty-one articles were evaluated for eligibility after abstract and title screening. Finally, 11 studies were included in the systematic review [9,14,15,16,17,18,19,20,21,22,23]. No additional studies were identified by checking the reference lists.

All studies were assessed as good or fair quality according to the Newcastle–Ottawa Scale for cohort studies (Table 1), and all were retrospective series.

In 11 studies, a total of 1207 patients were treated with definitive RTCT for advanced cervical cancer, and 601 patients underwent salvage surgery for persistent or recurrent disease after RTCT treatment: 369 (61.4%) hysterectomy (extrafascial or radical) and 232 (38.6%) pelvic exenteration (anterior, posterior, or total) (Table 2).

Table 2 illustrates the RTCT treatment administered to the patients in the different studies. In one study, there are no data about the dose of RTCT [16]. In two studies [15,16], the concomitant chemotherapy was cisplatin and 5-fluorouracil with respect to cisplatin administered in the other more recent studies included in the present review. Regarding the histopathological data, more than 80% of the patients had squamous cervical carcinoma (Table 2).

### 3.2. Pathological Findings

Overall, 439 (73%) patients had histologically confirmed residual disease in the salvage surgical specimen (Table 3). In six studies, the positive pathology after SS was 100%, establishing these series’ highly strict selection criteria [9,14,15,17,18,21].

All studies except one reported surgical margins analysis. In total, 109 patients had no complete R0 resection (with an overall range of 0–43% of the patients). Pervin et al. was the only series to show no positive margins at the final pathological evaluation (Table 3) [21]. Contrarily, Topuz et al. documented a higher rate of positive margins (43.5%) after SS, but in this series, 17 patients (74%) had undergone a simple hysterectomy [22].

Regarding the surgical procedure, about 70% (425/601) of the patients included in the studies underwent pelvic and/or aortic lymphadenectomy. Patients with at least one metastatic lymph node totaled 139 (32.7% of lymphadenectomy), with an overall range of 0–50%. Exclusively in one series, the authors reported no metastatic lymph nodes in 40 patients who underwent pelvic and para-aortic lymphadenectomy (Table 3) [21]. Interestingly, Gosset et al. showed that all patients with positive lymph nodes had lymphovascular space invasion (LVSI) [20].

### 3.3. Survival

The overall median follow-up was 38 months (range 13–93), based on the median of eight studies [9,14,16,18,19,21,22,23]. The recurrence rate was 31.6% (175/554 patients underwent SS with available recurrence data), with an overall DFS range of 1–114 months (Table 3).

The rate of death was 40% (229/572 patients underwent SS with available OS data), with an overall median OS of 32 months (range 9–239) (Table 3).

Interestingly, Nijhuis et al. showed that patients with vital tumor cells in their post-RTCT biopsy samples who underwent SS had a better OS than patients who did not receive adjuvant surgery (log-rank test, 7.22, *p* = 0.0072) [15].

Regarding SS radicality, Boers et al. reported that more radical surgery was not associated with improved disease-specific survival (HR, 0.84; 95% CI, 0.20–3.46; *p* = 0.81) [16]. Conversely, in the study of Pervin et al., all recurrences occurred in the extrafascial hysterectomy group [21].

Considering only the two studies that included exclusively pelvic exenteration, Chiantera et al. [17] reported a 5-year OS of 39% in the persistent disease group, and Stanca et al. [23] reported a 5-year OS of 48.7%.

A detailed report of oncological outcomes is shown in Table 3.

### 3.4. Postoperative Complications

All but one study reported severe (grade > 2) postoperative complications (Table 3). Ten studies reported an overall rate of severe complications of 29.8% (range 5–57.5%) [9,14,16,17,18,19,20,21,22,23]. Pervin et al. [21] documented the lower rate (5%) of severe postoperative complications of the selected studies: two patients had ureteral stenosis (grade 3) and both of them were in the group of radical hysterectomy. Touboul et al. [9] reported 25% of patients as having postoperative complications after hysterectomy (37/150), but this rate also includes grade 2 complications. The most frequent complication was lymphocyst (15%) and lymphedema (16%) [9].

Topuz et al. reported a complication rate of 10/17 for simple hysterectomy, and 2/6 for radical surgery [22]. Conversely, Boers et al. reported more treatment-associated morbidity in more radical surgery: grade 3 complications increased from 0% to 29% (*p* = 0.02) [16].

Interestingly, regarding the relationship between the rate of complication and the period from the last radiotherapy until surgery, Topuz et al. reported a lower rate of grade 3 complications in patients within two months compared to patients after two months (33.3% vs. 64.3%, respectively) [22].

## 4. Discussion

This systematic review analyzed the perioperative and survival outcomes of LACC patients who underwent RTCT and salvage surgery in eleven retrospective studies [9,17,18,19,20,21,22,23]. After a median follow-up of 38 months, the overall RR was about 32% with an overall death rate of 40% after hysterectomy or pelvic exenteration with or without lymphadenectomy. The risk of severe postoperative complications was 30%.

Limitations on this topic are the lack of sufficient knowledge due to the small patient population, the heterogeneity of the treatment between the clinical studies worldwide (e.g., radiation dose, use of brachytherapy, incomplete primary therapy), the selection of these patients after definitive RTCT (with or without biopsy), and the sensibility/specificity of imaging in this setting of patients.

To date, there are no retrospective, prospective, or randomized studies comparing the oncological outcomes of the two surgical approaches. Moreover, no comparable studies are available regarding the surgical versus the chemotherapy or wait-and-see follow-up in this setting of patients with persistent disease after RTCT treatment.

To study the effect of surgery, patients with residual disease should be identified because SS had no oncological benefit in patients with complete response after RTCT treatment [11], while also considering the high risk of severe postoperative complications that is about 30% which in line with other previous published series [24,25,26,27,28]. This treatment-related morbidity can decrease patients’ quality of life and influence the survival of patients with cervical cancer [29,30].

In the selected case, the advantage of SS could be doubled. In a retrospective cohort study, locoregional control was more often obtained in patients who underwent an operation (7 of 13) than in patients who were not selected for salvage surgery (0 of 8 patients) (*p* < 0.05) [15].

Furthermore, the SS after RTCT might give survival advice by providing relevant prognostic parameters (i.e., the pathologic assessment of residual disease in primary and lymph node sites) [31]. Indeed, the complete response has been associated with better survival in terms of DFS and OS. However, only a few studies present an acceptable sample size [32,33].

On the other hand, it is essential to evaluate the extent of persistent disease preoperatively for proper surgical management, including in the decision-making process the independent prognostic factors, such as the size of the tumor (more or less than 5 cm), histologic type, lymph nodes metastasis, LVSI, and the possibility to achieve free surgical margins [34].

The only modifiable parameter that the surgeon could influence is the achievement of surgical margins with more radical surgery. Clear margins are required for curative salvage surgery. Indeed, positive surgical margins are the major significant and independent prognostic factor associated with decreased survival of patients [17]. Postoperative survival at two years drops from 55.2% with uninvolved margins to 10.2% with positive margins (*p* = 0.0057) [35].

Frozen sectioning is a potential tool for improving survival and the benefit of surgery by to achieving disease free surgical margins. To our knowledge, there is no data about the surgical specimen’s intraoperative analysis, hence future studies are needed.

The burning question is how we can identify patients with residual disease eligible for potential benefit surgery and avoid pointless or detrimental salvage surgery. Unfortunately, the accuracy of treatment response evaluation before surgery is poor [20].

The post-RTCT evaluation should be based on a multidisciplinary algorithm including radiological, nuclear medical doctor, clinical, and if necessary, histopathological assessment to make the correct interpretation and avoid possible pitfalls in the decision-making process.

One of the major issues is the radiological evaluation of bladder and/or bowel residual infiltration in case of persistent disease after RTCT in FIGO stage IVA patients. This information is relevant for modulating the radicality of surgery (hysterectomy vs. PE) in patient candidates for SS because the main prognostic factor is the free surgical margins.

For the positive predictive value (PPV) of MRI for bladder invasion, studies show highly discrepant values ranging from 7% to 100% [36,37,38,39].

However, there are no studies about the invasion in the setting of post-RTCT patients.

As a consequence of RTCT, the cervix and adjacent soft tissues undergo fibrotic changes, appearing diffusely hypointense on T2WI, and the increased SI from RTCT-induced edema, flogosis, and necrosis may persist for up to 6 months, mimicking residual tumor [39].

Consequently, the interpretation of the post-treatment pictures of the irradiated pelvic disease is one of the major challenges for the expert radiologist.

There is an emerging need for functional imaging to overcome the high rate of MRI false-positive results, and T2-W and diffusion-weighted imaging (DWI) represent the mainstay of diagnostic sequences [40,41].

The use of DCE-MRI in post-RTCT cervical cancer could help select patients at risk of progression or early recurrence and, therefore, super select patient candidates for salvage surgery [42,43].

Furthermore, a new scenario is opening in the era of personalized medicine, and radiomics represents one of the more remarkable improvements in discovering new non-invasive biomarkers.

Radiomics converts medical images into minable data and provides insights into diseases that are imperceptible to human observers [44,45].

Recently, Autorino et al. proposed a radiomic model to predict 2 years OS in LACC patients before or during RTCT, and it could be of added clinical value to provide guidance for clinicians in their decision-making process to adapt and tailor treatment [46]. However, prospective and multicentric larger trials are needed to validate this model and eventually translate it into clinical practice.

Another helpful approach to identify patients with the central residual disease early is the examination under anesthesia (EUA) with cervical biopsy 8 to 10 weeks after completion of RTCT [47].

Of note, the clinical examination has a low sensitivity (51%) and specificity in detecting residual disease (62%) [48]. In a study by Fanfani, EUA post-RTCT in 35 patients with FIGO stage III cancer revealed a residual disease rate of 8%, whereas the histopathological residual disease rate after hysterectomy was 45.6% [49].

Consequently, a turning point could be the personalized selection of patients who would benefit from salvage surgery and confirmation of residual disease with biopsy is pivotal before performing salvage surgery [50]. In a systematic review, Van Kol et al. showed that the women who underwent SS based on the residual disease on biopsy showed a higher rate of positive specimens at final pathology than the women treated with SS based on radiology findings alone [50].

However, in some cases of RTCT-treated patients, the lesion could be difficult to be biopsied with a higher rate of false negative results due to the fibrosis, the flogosis, and the retraction of the vaginal fornix after the treatment. In this case, a minimally invasive procedure such as a transvaginal ultrasound-guided biopsy may provide a conclusive histological diagnosis [51].

Moreover, the timing of the biopsy should be standardized. When biopsies are taken too early, cervical cancer is still going into regression due to the continuing effect of radiation therapy. This could lead to a higher percentage of false-positive residual disease. In this review, the lack of these data did not permit a specific analysis. To reduce unnecessary salvage surgery, post-CRT biopsies should be taken 12 to 16 weeks after completion of RTCT [52].

In patients with positive lymph nodes prior to CRT treatment, selection with post-RTCT biopsy remains a point of discussion, and the survival impact of lymph nodal metastasis should be investigated. In a recent retrospective series, Mabuchi et al. showed that the presence, not number or location, of lymph node metastasis was an independent poor prognostic factor for postoperative recurrence (hazard ratio (HR) 5.36; 95% CI 1.41–6.66; *p* = 0.0020) in patients who developed locally recurrent or persistent cervical cancer treated with salvage hysterectomy plus lymphadenectomy [53].

## 5. Strengths and Limitations

The strengths of this study are the comprehensive search strategy and the study design, ensuring that the assessment of the articles and the data extraction were performed independently by two reviewers. The quality assessment of this review shows a good or fair quality of the included articles, indicating a low methodological bias risk. This review provides relevant information about salvage surgery after RTCT treatment.

However, the evaluation of data has limitations. Based on the results of these 11 studies, the limit of this review is due to the high heterogeneity of the few studies published so far on this topic. Nevertheless, it represents an attempt to summarize all the evidence published in the literature on this tool. Moreover, given the studies’ broad inclusion period, the RTCT treatment was different in all patients, and the diagnostic techniques have been changed and improved in the last few years. Furthermore, the median follow-up was wide in the included studies, with a range of 13 to 93 months, and it could determine the survival analysis. Last, not all studies performed the same type of radicality; this could influence the completeness of the surgery and the risk of complications.

## 6. Conclusions

In conclusion, no retrospective or prospective studies analyze the oncological outcomes by directly comparing the hysterectomy to pelvic exenteration in cases of persistent disease after definitive RTCT in LACC patients.

This is a “gray scenario” in which a multidisciplinary approach is needed; furthermore, the timing of radiological assessment and biopsy should be standardized to detect and localize the residual tumor, which is essential for the timely escalation of therapy to salvage options. 

In the era of individualized medicine, a thorough selection of patients and extensive counseling on survival benefits and complication risks are recommended.

The oncological gynecologist may tailor the radicality of surgery using all the preoperative and intraoperative tools to achieve free surgical margins that, to date, remain the only modifiable prognostic factor.

## Figures and Tables

**Figure 1 medicina-59-00192-f001:**
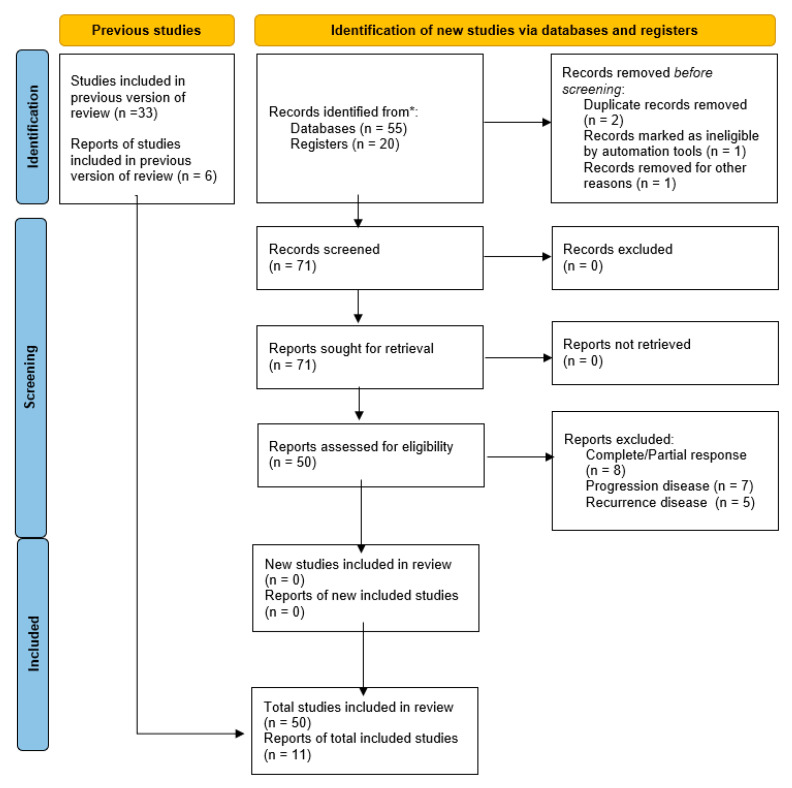
Flowchart of the selection procedure for systematic review. * Consider, if feasible to do so, reporting the number of records identified from each database or register searched (rather than the total number across all databases/registers).

**Table 1 medicina-59-00192-t001:** Newcastle–Ottawa Scale for cohort studies included in the analysis [9,14,15,16,17,18,19,20,21,22,23].

Author	Selection				Comparability	Outcome			
	Representativeness of the Exposed Cohort ^a^	Selection of the Non-Exposed Cohort	Ascertainment of Exposure ^b^	Demonstration That Outcome of Interest Was not Present at Start of Study	Comparability of Cohorts	Assessment of Outcome ^c^	FUP ^d^	Adequacy of FUP	Quality
Azria (2005) [14]		NA			NA		22 (range 1–37)	1 patient lost to FUP	**Good**
Nijhuis (2006) [15]							62.4 (range 46.8–105.6)	No lost to FUP	**Good**
Boers (2014) [16]							27.6 (IQR, 15.6–56.4)	No lost to FUP	**Good**
Chiantera (2014) [17]							68 (range 47–94)	No lost to FUP	**Good**
Mabuchi (2017) [18]							41.5 (mean)	No statement about lost to FUP	**Good**
Platt (2018) [19]		NA			NA		31 (range 12–60)	No lost to FUP	**Good**
Gosset (2019) [20]		NA			NA		No statement about FUP	No statement about lost to FUP	**Fair**
Pervin (2019) [21]		NA			NA		93.6 (mean)	No statement about lost to FUP	**Fair**
Topuz (2020) [22]		NA					20 (range 6–118)	1 patient lost to FUP	**Good**
Touboul (2020) [9]							43.2 (range 2.4–127.2)	4 patients lost to FUP	**Good**
Stanca (2022) [23]		NA			NA		44.5 (1–88)		**Good**

^a^ Representativeness of the exposed cohort: all included studies representative of women with residual cervical cancer after chemoradiation therapy treated with salvage surgery. ^b^ Ascertainment of exposure: all with database or medical records. ^c^ Assessment of outcome: all with medical records. ^d^ FUP (expressed in months) ≥ 12 months was assessed as long enough for outcomes to occur. Abbreviations: NA = not applicable; FUP = follow-up period; IQR = interquartile range.

**Table 2 medicina-59-00192-t002:** Characteristics of studies included in the analysis [9,14,15,16,17,18,19,20,21,22,23].

Authors (Year of Publication)	Total Number of Pts Included in the Study (Pts with Persistent Disease)	FIGO Stage at Diagnosis	Histology	RTCT Treatment	Total Number of Patients with Salvage Surgery (H and PE)	Lymphadenectomy(P and AO)	Median Tumor Size, mm (Range)
Azria (2005) [14]	10 (10)	IIA = 2 IIB = 8	SCC = 8 AC = 2	EBRT 45 Gy with concomitant cisplatin 40 mg/m^2^) with 15 GyBRT	10 (H = 9) (PE = 1)	P = 8 AO = 8	60 (40–90)
Nijhuis (2006) [15]	165 (21)	NA	NA	EBRT 45 Gy with concomitant carboplatin and 5-FU (before 1999) and cisplatin 40 mg/m^2^ (after 1999) with 35 Gy BRT	13 (H = 12) (PE = 1)	0	NA
Boers (2014) [16]	491 (84)	IB1 = 4 IB2 = 16 IIA = 8 IIB = 29 IIIA = 2 IIIB = 2	SCC = 35 AC = 20 ACS = 2	EBRT 45 Gy in fractions of 1.8 Gy, from 1994 additional BRT total dose 34.8 Gy. Before 1999 carboplatin (300 mg/m^2^) and 5 FU (600 mg/m^2^). After 1999 cisplatin 40 mg/m^2^)	61 (H = 56) (PE = 5)	0	NA
Chiantera (2014) [17]	167 (34)	NA	SCC = 144 AC = 22 Other = 1	NA	167 (PE = 167)	P = 83 PA = 46	NA
Mabuchi (2017) [18]	51 (34)	IB2-IIA = 10 IIB-IVA = 39 IVB = 2	SCC = 29 AC = 19 Other = 3	EBRT 45 Gy with/without concomitant weekly cisplatin 40 mg/m^2^ with 3 doses of BRT	46 (H = 37) (PE = 9)	P and/or PA = 36	30 (5–70)
Platt (2018) [19]	15 (15)	IB2 = 2 IIA = 1 IIB = 9 IIIB = 3	SCC = 8 AC = 7	EBRT 45 Gy with concomitant weekly cisplatin 40 mg/m^2^ with 3 doses of BRT	15 (H = 15)	0	44 (23–53)
Gosset (2019) [20]	31 (29)	IB2 = 8 IIA = 2 IIB = 18 III = 1	SCC = 22 AC = 7	EBRT 45–50.4 Gy in 25–28 fractions of 1.8 Gy over 5 weeks with concomitant cisplatin (40 mg/m^2^) with 15 Gy BRT	29 (H = 29)	P = 1 PA = 19 P and PA = 1	19 (6–40)
Pervin (2019) [21]	55 (40)	IIB = 25 IIIB = 7 At least IIB = 8	SCC = 32 AC = 7 ACS = 1	EBRT 50 Gy in 25fractions of 2 Gy for25 days with or without3 × 7 Gy BT. 23 patients received 3 doses cisplatin	40 (H = 40)	P and PA = 40	>2 cm (30%) <2 cm (70%)
Topuz (2020) [22]	25 (23)	At least IB2 = 23	SCC = 16 AC = 7	EBRT, 1.8–2 Gy per fraction, Total 45–50 Gy with cisplatin 40 mg/m^2^/week and BT 5 Gy once weekly for 5 weeks	23 (H = 21) (PE = 2)	0	NA
Touboul (2020) [9]	150 (78)	IB2 = 48 II = 91 III = 10 IV = 1	SCC = 108 AC = 26 Other = 16	EBRT 45 Gy with concomitant weekly cisplatin 40 mg/m^2^ with BRT 15 Gy	150 (H = 150)	P = 5 PA = 82 P & PA = 49	NA
Stanca (2022) [23]	47 (5)	NA	SCC = 40 AC = 7	EBRT 45 Gy with concomitant weekly cisplatin 40 mg/m^2^ with BRT 15 Gy	47 (PE = 47)	P = 47	<4 cm (51.1%) ≥4 cm (48.9%)

Abbreviations: NA = not available; SCC = squamous-cell carcinoma; AC = adenocarcinoma; ACS = adenosquamous carcinoma; EBRT = external-beam radiotherapy; BRT = brachytherapy; fluorouracil (FU); H = hysterectomy; PE = pelvic exenteration; P = pelvic; PA = para-aortic.

**Table 3 medicina-59-00192-t003:** Outcomes of studies included in the analysis [9,14,15,16,17,18,19,20,21,22,23].

Authors (Year of Publication)	Positive Pathology after Salvage Surgery (%)	Positive Margins (%)	Patients with at Least One Metastatic Lymph Node (%)	N severe Complications Grade ≥ 3 (%)	Follow-Up Period Median Months (Range)	Recurrence (%)	Median DFS Months (Range)	Death (%)	Median OS Months (Range)
Azria (2005) [14]	10 (100)	1 (10)	5 (50)	4 (40)	22 (1–37)	7 (70)	18 (6–36)	4 (40)	24 (9–37)
Nijhuis (2006) [15]	13 (100)	2 (15.4)	0	NA	NA	8 (61.5)	NA	8 (61.5)	48 (11–105)
Boers (2014) [16]	44 (72)	8 (13)	0	14 (23)	27.6	31 (50)	NA	29 (47.5)	NA
Chiantera (2014) [17]	167 (100)	46 (27.5)	49 (23)	58 (34.7)	NA	41 (33.9)	13.4 (1.4–114)	99 (59.3)	19 (15–239)
Mabuchi (2017) [18]	51 (100)	14 (27.4)	20 (39.2)	11 (21.6)	41.5	23 (45.1)	23.3	19 (37.3)	29
Platt (2018) [19]	4 (26.7)	NA	0	3 (20)	13 (12–60)	3 (20)	n.r. (10–31)	1 (6.7)	n.r.
Gosset (2019) [20]	14 (48.3)	2 (6.9)	3 (10.3)	7 (24)	NA	3 (10.3)	n.r. (9–43)	NA	NA
Pervin (2019) [21]	40 (100)	0	0	2 (5)	93.6 (60–108)	4 (10)	NA	1 (2.5)	NA
Topuz (2020) [22]	18 (78.2)	10 (43.5)	0	12 (52.2)	20 (6–118)	14 (60.8)	15 (6–23)	9 (39.1)	20 (in positive margins) 36 (in negative margins)
Touboul (2020) [9]	78 (52)	9 (7)	47 (31.3)	37 (25) *	43 (2–127)	41 (27)	5-ys = 66%	37 (24.7)	NA (5 ys = 71%)
Stanca (2022) [23]	47 (100)	17 (36)	15 (31.9)	27 (57.5)	44.5 (1–118)	NA	NA	22 (46.8)	49.4 (5 ys = 48.7%)

* grade ≥ 2. Abbreviations: N = number; DFS = disease-free survival; OS = overall survival; n.r. = not reached; NA = not available; ys= years.

## Data Availability

Not applicable.

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
