# Peer review of "Assessment of Salvage Surgery in Persistent Cervical Cancer after Definitive Radiochemotherapy: A Systematic Review"

_medicina, 2023, doi:10.3390/medicina59020192_

Round 1

Reviewer 1 Report

This systematic review evaluates the role of salvage surgery for residual disease after CRT for LACC. I have the following comments:

1.       In the introduction section the authors should add the definitions of residual and recurrent disease (are the definitions uniform with regards to the time of relapse?)

2.       The authors briefly mention international guidelines and state that CRT and PE should not be combined. What are the indications for PE in LACC?

3.       I would suggest the discussion section to be shortened.

4.       Lines 304-6: the authors mention examination under anaesthesia and provide no reference. Please add the reference or omit this sentence.

Author Response

This systematic review evaluates the role of salvage surgery for residual disease after CRT for LACC. I have the following comments:

  1. In the introduction section, the authors should add the definitions of residual and recurrent disease (are the definitions uniform regarding the time of relapse?)

Response 1: Thank you for the comment. As the reviewer suggested, we better specified the residual and recurrent disease in the introduction section.

See revised manuscript lines 93-97

  1. The authors briefly mention international guidelines and state that CRT and PE should not be combined. What are the indications for PE in LACC?

Response 2: Thank you for the comment. According to NCCN guidelines, the standard treatment in LACC patie is the exclusive concurrent chemoradiation therapy with a high pathological response after 6 months (about 70%) [references 3-5]. The combination of a PE and RTCT should be avoided given the high morbidity rate (more than 30% of major postoperative complications) [see reference 18]

Currently, the most common indication for PE is central recurrent cervical cancer after chemoradiation, in absence of other curative option, remains. Whereas most surgeons consider intraperitoneal tumor spread and distant metastases absolute contraindications to PE, lymph node involvement or large tumor size, despite having both a negative impact on overall survival (OS), are not unanimously regarded as absolute exclusion criteria for PE at recurrence.

As suggested by the Reviewer we better specified in the manuscript. See revised manuscript, line 88

  1. I would suggest the discussion section to be shortened.

Response 3: As suggested by the Reviewer, we deleted some sentences of the discussion, particularly the paragraph about the radiological assessment.

See the revised manuscript.

  1. Lines 304-6: the authors mention examination under anesthesia and provide no reference. Please add the reference or omit this sentence.

Response 4: As correctly suggested by the Reviewer, we added the reference.

Please, see reference 49 of the revised manuscript.

Reviewer 2 Report

I think that this report is valid and interesting, because these data could be presented when the patients selected treatment methods. So I have only one question.

In Table 2 and 3, in the reports of Nijhuis et al. and Boers et al., the patients with salvage surgery were minority? If so, the results in these reports may become poor compared with those in other hospitals.

Author Response

I think that this report is valid and interesting, because these data could be presented when the patients selected treatment methods. So I have only one question.

In Table 2 and 3, in the reports of Nijhuis et al. and Boers et al., the patients with salvage surgery were minority? If so, the results in these reports may become poor compared with those in other hospitals.

RESPONSE: Thank you for the revision. As we reported in the limitation paragraph of the discussion, there is a heterogeneity in the few studies published on this topic, which is a challenging treatment for oncological gynecologists.

The studies Nijhuis et al. and Boers et al. are two monocentric retrospective series in this field that fulfilled the inclusion criteria of this systematic review. Both studies included a large number of LACC patients treated with RTCT. However, they correctly selected for the analysis only patients with persistent disease after RTCT (12% and 17%, in the Nijhuis and Boers studies, respectively). In particular, Boers et al. included 61 patients who underwent salvage surgery in the final analysis, and it is one of the larger series to analyze the oncological outcome of this setting of patients.